

# Does long-term fire suppression impact leaf litter breakdown and aquatic invertebrate colonization in pine flatwoods wetlands?

Houston C. Chandler[1,2], J. Checo Colón-Gaud[3], Thomas A. Gorman[1,4], Khalil Carson[3,5] and Carola A. Haas[1]

[1] Department of Fish and Wildlife Conservation, Virginia Polytechnic Institute and State University (Virginia Tech), Blacksburg, VA, United States of America

[2] The Orianne Society, Tiger, GA, United States of America

[3] Department of Biology, Georgia Southern University, Statesboro, GA, United States of America

[4] Aquatic Resources Division, Washington State Department of Natural Resources, Olympia, WA, United States of America

[5] Biological and Environmental Sciences Department, Troy University, Troy, AL, United States of America

Corresponding author
Houston C. Chandler,
houstonc@vt.edu

## ABSTRACT

Ephemeral wetlands are commonly embedded within pine uplands of the southeastern United States. These wetlands support diverse communities but have often been degraded by a lack of growing-season fires that historically maintained the vegetation structure. In the absence of fire, wetlands develop a dense mid-story of woody vegetation that increases canopy cover and decreases the amount of herbaceous vegetation. To understand how reduced fire frequency impacts wetland processes, we measured leaf litter breakdown rates and invertebrate communities using three common plant species (Longleaf Pine (*Pinus palustris*), Pineland Threeawn Grass (*Aristida stricta*), and Black Gum (*Nyssa sylvatica*)) that occur in pine flatwoods wetlands located on Eglin Air Force Base, Florida. We also tested whether or not the overall habitat type within a wetland (fire maintained or fire suppressed) affected these processes. We placed leaf packs containing 15.0 g of dried leaf litter from each species in both fire-maintained and fire-suppressed sections of three wetlands, removing them after 103–104 days submerged in the wetland. The amount of leaf litter remaining at the end of the study varied across species (*N. sylvatica* = 7.97 ± 0.17 g, *A. stricta* = 11.84 ± 0.06 g, and *P. palustris* = 11.37 ± 0.07 g (mean ± SE)) and was greater in fire-maintained habitat (leaf type: $F_{2,45} = 437.2$, $P < 0.001$; habitat type: $F_{1,45} = 4.6$, $P = 0.037$). We identified an average of 260 ± 33.5 (SE) invertebrates per leaf pack (range: 19–1,283), and the most abundant taxonomic groups were Cladocera, Isopoda, Acariformes, and Diptera. Invertebrate relative abundance varied significantly among litter species (approximately 39.9 ± 9.4 invertebrates per gram of leaf litter remaining in *N. sylvatica* leaf packs, 27.2 ± 5.3 invertebrates per gram of *A. stricta*, and 14.6 ± 3.1 invertebrates per gram of *P. palustris* (mean ± SE)) but not habitat type. However, both habitat (*pseudo*-$F_{1,49} = 4.30$, $P = 0.003$) and leaf litter type (*pseudo*-$F_{2,49} = 3.62$, $P = 0.001$) had a significant effect on invertebrate community composition. Finally, this work was part of ongoing projects focusing on the conservation of the critically imperiled Reticulated Flatwoods Salamander (*Ambystoma bishopi*), which breeds exclusively in pine flatwoods wetlands, and we examined the results as they relate to potential prey items for larval flatwoods

salamanders. Overall, our results suggest that the vegetation changes associated with a lack of growing-season fires can impact both invertebrate communities and leaf litter breakdown.

# INTRODUCTION

Natural disturbances, events that disrupt an ecosystem or change the physical environment, occur across a variety of both spatial and temporal scales and have historically played a critical role in shaping many ecosystems (*White & Pickett, 1985*; *White & Jentsch, 2001*; *Turner 2010*). However, anthropogenic activity has caused significant changes to individual disturbance events and to overall disturbance regimes (*Johnstone et al., 2016*; *Newman, 2019*). These changes are important because disturbances can promote biodiversity and habitat heterogeneity that would otherwise be lost (*Carlson et al., 1993*; *Townsend & Scarsbrook, 1997*; *Conway, Nadeau & Piest, 2010*). Thus, restoring ecologically important disturbances are often the target of active management programs that attempt to emulate natural processes that have been disrupted by anthropogenic activity (*Long, 2009*).

Wildfire is a widespread natural disturbance, impacting a wide variety of terrestrial and aquatic ecosystems (*Morgan et al., 2001*; *Bisson et al., 2003*; *Mouillot & Field, 2005*; *Butz, 2009*). In the southeastern United States, the Longleaf Pine (*Pinus palustris*) ecosystem historically covered an area of approximately 35 million hectares (*Frost, 1993*). Longleaf Pine forests are a classic example of a fire-adapted ecosystem, experiencing frequent low-intensity fires (*Henderson, 2006*; *Stambaugh, Guyette & Marshall, 2011*). These regular fires historically maintained vegetation structure (*e.g.*, a pine overstory with thick herbaceous vegetation on the forest floor) and reduced leaf litter and woody debris build up (*Brockway & Lewis, 1997*). In the absence of natural or prescribed fires, Longleaf Pine forests transition to closed-canopy systems with abundant hardwoods in the mid-story and decreased diversity and abundance of herbaceous vegetation on the forest floor (*Gilliam & Platt, 1999*; *Glitzenstein, Streng & Wade, 2003*).

Ephemeral wetlands are a common landscape feature in Longleaf Pine forests and are characterized by a regular wetting and drying cycle that is tied to annual variation in precipitation and evapotranspiration rates. These wetlands are frequently geographically isolated (*i.e.,* lacking a consistent surface water connection to other water bodies; (*Tiner, 2003*; *Cohen et al., 2016*) and commonly support abundant and diverse biotic communities that are dependent on relatively predictable periods of inundation (*e.g.*, *Golladay, Taylor & Palik, 1997*; *Kirkman et al., 1999*; *Erwin et al., 2016*). Furthermore, these wetlands are subject to vegetation shifts similar to other Longleaf Pine ecosystems that can occur both from fire suppression and from poorly timed (*e.g.*, during the winter months when wetlands are more likely to have standing water present) prescribed fires (*Kirkman, 1995*; *Bishop & Haas, 2005*). Vegetation shifts change aquatic systems by altering leaf litter inputs,

reducing the amount of structure available in the aquatic environment, and impacting other ecosystem processes (*Mulhouse et al., 2005*; *Hinman & Brewer, 2007*). Ultimately, these types of changes can impact the composition and abundance of aquatic communities (*Hornung & Foote, 2006*; *Chandler et al., 2015*).

Aquatic invertebrate communities are a critical component of ephemeral wetlands (*Batzer & Wissinger, 1996*; *McInerney et al., 2017*), functioning across multiple trophic levels by acting as both a prey base for other species and as predators in generally fishless environments (*Murkin & Wrubleski, 1988*; *Batzer & Wissinger, 1996*). Ephemeral wetlands can also support higher aquatic invertebrate biomass and differing community composition when compared to permanent wetlands with fish populations (*Zimmer et al., 2001*; *McInerney et al., 2017*). One of the important roles that invertebrates play in wetland ecosystems is contributing to leaf litter breakdown by consuming and physically breaking down leaf litter that falls into the wetland basin (*Fazi & Rossi, 2000*; *Gingerich, Panaccione & Anderson, 2015*). Breakdown rates can vary widely across different leaf litter species (*Leroy & Marks, 2006*), and litter inputs into the wetland can broadly impact both biotic and abiotic processes (*Stoler & Relyea, 2016*). The structure, nutrient content, and availability of individual leaf litter species can all impact the invertebrate community, with effects potentially transitioning to higher trophic levels (*Batzer & Palik, 2007*; *Stoler & Relyea, 2016*).

Here, we describe the results of a field experiment testing the effects of habitat type (*i.e.,* fire maintained *vs.* fire suppressed) on leaf litter breakdown and invertebrate communities in pine flatwoods wetlands. We assessed these environmental processes for three species of leaf litter: Longleaf Pine (*P. palustris*), Pineland Threeawn Grass, commonly referred to as wiregrass, (*Aristida stricta*), and Black Gum (*Nyssa sylvatica*). *Pinus palustris* and *A. stricta* are commonly found in fire-maintained wetlands, while *N. sylvatica* is mostly restricted to wetlands with reduced fire frequency or deeper portions of wetlands that are less likely to experience regular fires (*Chandler, 2015*). We predicted that fire-maintained sections of wetlands would support more abundant invertebrate communities, leading to higher breakdown rates. Furthermore, we predicted that *P. palustris* and *A. stricta* would breakdown more slowly than *N. sylvatica* because of lower surface area for invertebrate colonization. Finally, we assessed invertebrate communities overall and specifically focused on taxa that are important food sources for larval Reticulated Flatwoods Salamanders (*Ambystoma bishopi*), a US Federally endangered species (*U. S. Fish and Wildlife Service, 2009*). Larval flatwoods salamanders feed primarily on aquatic invertebrates (*e.g.,* Isopoda, Amphipoda, and Copepoda) and, while little is known about foraging behavior, opportunistic observations suggest that larvae forage in and around herbaceous vegetation as well as along benthic substrates (*Palis, 1996*; *Sekerak, Tanner & Palis, 1996*; *Whiles et al., 2004*).

## MATERIALS & METHODS

### Study sites

All field work was conducted on Eglin Air Force Base (Eglin) and access to field sites was approved by the US Fish and Wildlife Service and Jackson Guard (Eglin's Natural

Resources Division; Cooperative Agreement Number F14AC00068). Eglin is a large military installation covering over 188,459 ha of the Florida Panhandle's Gulf Coastal Plain. Consisting largely of sandhills and other upland habitat, Eglin also contains some of the best remaining examples of mesic pine flatwoods. These forests have sandy, poorly drained soils and ephemeral, geographically isolated wetlands embedded within the surrounding pine forest. Eglin has an extensive active habitat management program that routinely applies prescribed fire to pine forests across the installation. However, there have been persistent challenges associated with burning inside of wetland basins, and most pine flatwoods wetlands on the installation were either partially or completely degraded by long-term fire suppression and exclusion. Pine flatwoods wetlands on Eglin support diverse amphibian communities, including breeding populations of Reticulated Flatwoods Salamanders (*U. S. Fish and Wildlife Service, 2009*). During their aquatic larval phase, flatwoods salamanders depredate a variety of freshwater invertebrate groups (*Whiles et al., 2004*), serving as important predators in the wetlands that they inhabit.

## Study design

We collected leaf litter from the three species of interest (*P. palustris*, *A. stricta*, and *N. sylvatica*) in areas surrounding pine flatwoods wetlands on Eglin. We raked *P. palustris* needles from underneath large trees, collected *N. sylvatica* leaves from plants that had been manually removed from wetlands, and collected *A. stricta* material from standing dead stems at the end of the reproductive period. We dried all leaf litter at room temperature for approximately one week. We then filled coarse mesh bags (8-mm openings), allowing for invertebrate colonization, with 15.0 g of dried leaf litter from a single species.

We placed a total of 72 leaf packs across three pine flatwoods wetlands on Eglin that were part of long-term monitoring projects focused on wetland communities and management (*e.g.*, *Gorman, Haas & Himes, 2013*). We selected wetlands that had areas of both high (fire maintained) and low (fire suppressed) herbaceous vegetation cover. We note here that we chose wetlands based on their vegetation characteristics and not on a specific fire history. All wetlands are located in actively managed pine flatwoods on Eglin with similar overall management histories. Based on limited data, wetland basins included in this study experienced 2–3 prescribed fires from 2012–2015, and we documented fire effects 86% of the time in fire-maintained sections and 29% of the time in fire-suppressed sections (based on leaf pack locations). Furthermore, current conditions are the result of multiple management activities, including both prescribed fire and mechanical removal of woody vegetation (*Gorman, Haas & Himes, 2013*).

In each wetland, we deployed leaf packs along 10-m transects in both habitat types, placing one leaf pack of each plant species at both ends and in the middle of the transect (9 per transect and 18 per wetland). We separated individual leaf packs by approximately 30 cm perpendicular to the transect. We added leaf packs to wetlands on 10 and 11 November 2015 and secured them to the bottom of the wetland using metal gardening stakes. Finally, in one wetland, we doubled the number of leaf packs to examine the effects of time submerged on breakdown and invertebrate communities (logistical constraints

prevented this effort in all wetlands). We positioned these leaf packs adjacent to the same transect, placing leaf packs containing the same plant species next to one another.

Wetlands remained flooded for the entire duration of the study, and we removed leaf packs at two time periods. First, we collected the leaf packs from the additional replicate in the single wetland on 5 December 2015 (25 days submerged). All other leaf packs were removed from wetlands on 22 February 2016 (103–104 days submerged). Upon removal from the wetland, we immediately placed all leaf packs in 95% ethanol to stop further decomposition of leaf material and to preserve invertebrates until samples could be processed.

We washed all of the remaining leaf litter to separate out invertebrates using a 125 µm sieve. All remaining material from the leaf packs was then dried at 55 °C for at least 48 h. We then weighed this material to obtain dry mass (DM). Next, we ground samples into a fine powder by milling at 25,000 rpm for either 90 s (*N. sylvatica*) or 180 s (*P. palustris* and *A. stricta*) (IKA® Tube Mill 100 control). We burned ground samples in a muffle furnace at 500 °C for 1 h to obtain ash-free dry mass (AFDM). For the 25-day replicate, we burned the entire sample, while we burned approximately 1 g of the 103–104-day samples and calculated the AFDM from this subsample. We used the AFDM of each sample to calculate the processing coefficient (k) using the exponential decay model (*Petersen & Cummins, 1974*; *Maloney & Lamberti, 1995*; *Benfield, Fritz & Tiegs, 2017*). We used the estimated processing coefficients to calculate number of days it would take for 99% of the sample to breakdown. Finally, we identified all invertebrates to broad taxonomic groups that were easily identifiable (order in most cases).

In addition to leaf packs, we also quantified herbaceous vegetation cover and canopy cover as part of other ongoing projects. Herbaceous vegetation and canopy cover were measured at points along each wetland's longest axis, partially overlapping the locations with leaf packs (*Gorman, Haas & Himes, 2013*). We estimated the percent herbaceous vegetation cover using a 0.5 m × 0.2 m Daubenmire frame and the Daubenmire cover class scale (*Daubenmire, 1959*). We measured canopy cover by averaging the canopy cover recorded in each of the four cardinal directions with a convex spherical densiometer. All vegetation data were collected during the fall of 2014, and for this assessment, we included the two sampling locations that were closest to the leaf packs.

## Statistical analyses

Prior to all analyses, we excluded all results from one 104-day leaf pack (*A. stricta*, fire-suppressed habitat) and the invertebrate results from one 25-day leaf pack (*P. palustris*, fire-maintained habitat) because these data were not consistent with the other results. Furthermore, we only included data from leaf packs submerged for 103–104 days in the following analyses. We tested for an effect of leaf litter species and habitat type (fire suppressed or maintained) on leaf litter breakdown and invertebrate abundance using a series of linear mixed effects models (LMM). First, we fit a mixed effects model to test for the effects of litter species and habitat type on the amount of leaf litter remaining at the end of the experiment. We treated the wetland as a random effect to account for non-independence in the leaf packs collected from the same wetland. Leaf litter species,

habitat type, and their interaction were included in the model as fixed effects. Second, we fit two similar models using the total invertebrate abundance and the abundance of groups that make up the primary components of the larval flatwoods salamander diet. We defined these groups as Isopoda, Amphipoda, and Copepoda, which make up approximately 65% of prey items found in larval flatwoods salamander stomachs (*Whiles et al., 2004*). Both measures of invertebrate abundance were standardized by the mass remaining in their respective leaf pack prior to analyses. Invertebrate models also contained leaf species, habitat type, and their interaction as fixed effects and wetland as a random effect. We verified that the assumptions of linearity, normality, and homogeneity of the residuals were met in all three models using diagnostic plots. Finally, we performed pairwise comparisons using Tukey's HSD when tests on main effects indicated a significant difference among leaf litter species.

To further examine the composition of invertebrate communities, we visualized the community data using a non-metric multidimensional scaling (NMDS) plot. We graphed community data using Bray–Curtis dissimilarities and verified that the Stress statistic for the NMDS was less than 0.2 (*Clarke, 1993*). We tested for differences in community composition among leaf litter species and habitat type using a distance-based permutational multi-variate analysis of variance (PERMANOVA; *Anderson & Walsh, 2013*). We conducted the PERMANOVA using Bray–Curtis dissimilarity indices and 999 permutations. The individual wetland was treated as a block in this analysis, and we tested for significance using the marginal effects. We also tested for differences in group dispersions (variance) following *Anderson (2006)*. All statistical analyses were performed in R (*R Core Team, 2020*), and NMDS, PERMANOVA, and tests for dispersion were available in the *vegan* package (*Oksanen et al., 2019*). Mixed models were fit using the *lme4* package (*Bates, Maechler & Walker, 2015*), and pairwise comparisons were performed using the *emmeans* package (*Lenth, 2021*).

## RESULTS

Across all three leaf species and both habitat types, the rate of mass loss was highest during the first 25 days after being submerged (Fig. 1). After 103–104 days submerged, *N. sylvatica* had the fastest mean k rate (−0.0062 ± 0.0002 (SE)) followed by *P. palustris* (−0.0027 ± 0.00006 (SE)) and *A. stricta* (−0.0023 ± 0.00005 (SE)). There was no interaction effect between habitat and leaf type on the amount of leaf litter remaining after 103–104 days in the wetland (LMM: $F_{2,45} = 2.0$, $P = 0.15$). However, both habitat (LMM: $F_{1,45} = 4.6$, $P = 0.037$) and leaf type (LMM: $F_{2,45} = 437.2$, $P < 0.001$) significantly impacted leaf litter breakdown after 104 days (Fig. 1; Table 1, Table 2). *Nyssa sylvatica* (weight remaining = 7.97 ± 0.17 g (mean ± SE)) leaves broke down faster than both *P. palustris* (weight remaining = 11.37 ± 0.07 g (mean ± SE)) ($P < 0.0001$) and *A. stricta* (weight remaining: = 11.84 ± 0.06 g (mean ± SE)) ($P < 0.0001$). Breakdown was slower, on average, in fire-suppressed sections of wetlands. However, the effect was small compared to the differences between leaf types and was not consistent across wetlands (Table 1).

We identified a total of 2,677 invertebrates in leaf packs (mean: 158 per pack, range: 14–379, SE: 32.2) collected after 25 days submerged and 14,062 invertebrates in leaf

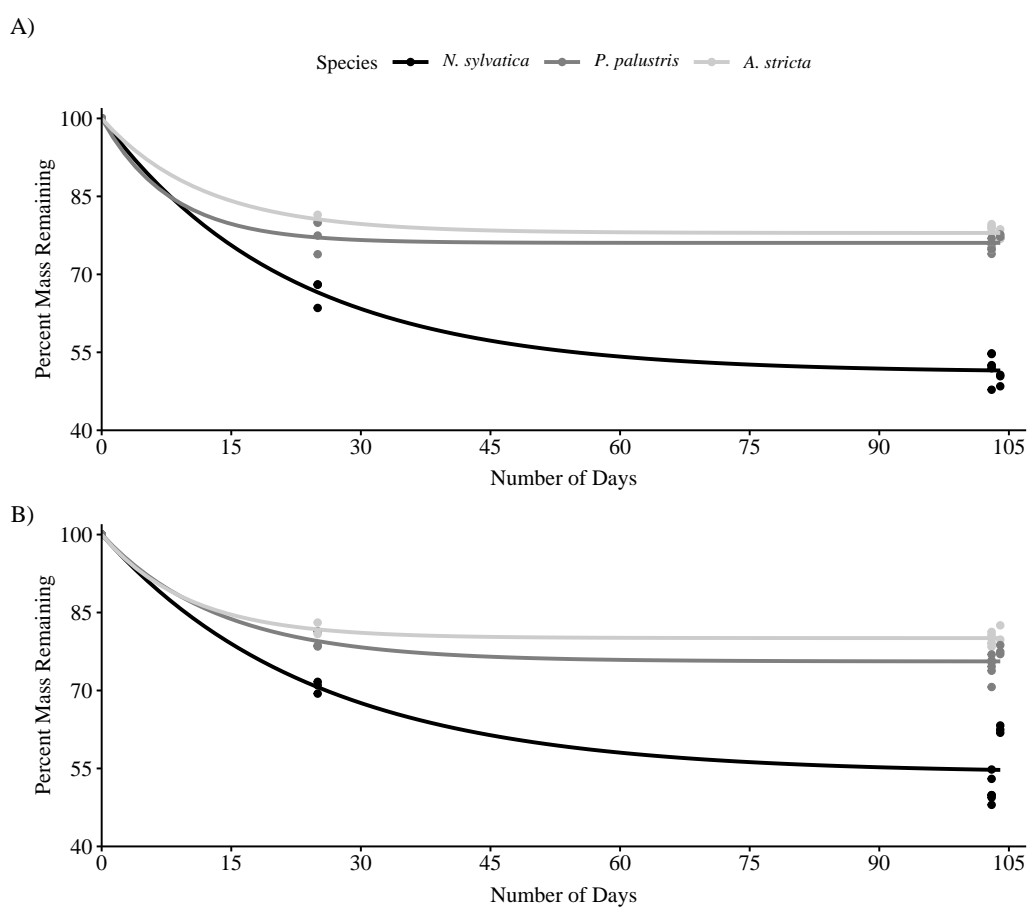

**Figure 1** **Mass loss for Longleaf Pine (*Pinus palustris*), Pineland Threeawn Grass (*Aristida stricta*), and Black Gum (*Nyssa sylvatica*) measured in three pine flatwoods wetlands on Eglin Air Force Base.** (A) Mass loss from fire-maintained habitats. (B) Mass loss from fire-suppressed habitats. Solid lines represent the non-linear least squares fit for each group, approximating the rate of mass loss over time using 25-day leaf packs collected in a single wetland.

packs (mean: 260 per pack, range: 19–1,283, SE: 33.5) collected after 103–104 days submerged. The most abundant invertebrate groups were Cladocera, Isopoda, Acariformes, and Diptera, accounting for approximately 93% of all invertebrates identified across all samples (Table 3). For leaf packs submerged 103–104 days, there was no interaction effect between litter species and habitat type on the total invertebrate relative abundance (LMM: $F_{2,45} = 0.52$, $P = 0.60$) or on the relative abundance of preferred flatwoods salamander prey items (LMM: $F_{2,45} = 0.12$, $P = 0.88$). Leaf type but not habitat type affected both overall invertebrate relative abundance (LMM: Leaf: $F_{2,45} = 3.98$, $P = 0.03$; Habitat: $F_{1,45} = 0.70$, $P = 0.41$) and the relative abundance of flatwoods salamander prey items (LMM: Leaf: $F_{2,45} = 13.4$, $P = 0.00003$; Habitat: $F_{1,45} = 0.40$, $P = 0.53$; Table 2). *Nyssa sylvatica* leaf packs (39.9 ± 9.4 (mean ± SE) invertebrates per gram of leaf litter remaining) had higher total invertebrate relative abundance than *P. palustris* (14.6 ± 3.1 (mean ± SE) invertebrates per gram of leaf litter remaining; Tukey's HSD: $t_{45} = 2.8$, $P = 0.007$) but not *A. stricta*

**Table 1 Leaf litter breakdown metrics recorded in pine flatwoods wetlands on Eglin Air Force Base, Florida.** Breakdown metrics for three plant species (Longleaf Pine (*Pinus palustris*), Pineland Threeawn Grass (*Aristida stricta*), and Black Gum (*Nyssa sylvatica*)) after 103–104 days submerged. Leaf packs were placed in sections of wetlands with vegetation characteristics indicative of fire-maintained and fire-suppressed habitat. Values represent means ± standard deviations from three leaf packs of each species. The processing coefficient (k) was estimated using an exponential decay model.

| | % Litter remaining | $k$ | Days to 99% decomposition |
|---|---|---|---|
| **Pond 53** | | | |
| Fire maintained | | | |
| *A. stricta* | 0.78 ± 0.01 | −0.0024 ± 0.0001 | 1898 ± 91 |
| *P. palustris* | 0.77 ± 0.004 | −0.0025 ± 0.0001 | 1857 ± 37 |
| *N. sylvatica* | 0.50 ± 0.01 | −0.0067 ± 0.0002 | 688 ± 24 |
| Fire suppressed | | | |
| *A. stricta* | 0.81 ± 0.02 | −0.0020 ± 0.0002 | 2305 ± 261 |
| *P. palustris* | 0.78 ± 0.01 | −0.0024 ± 0.0001 | 1906 ± 87 |
| *N. sylvatica* | 0.63 ± 0.01 | −0.0045 ± 0.0001 | 1020 ± 24 |
| **Pond 16** | | | |
| Fire maintained | | | |
| *A. stricta* | 0.77 ± 0.02 | −0.0025 ± 0.0003 | 1843 ± 187 |
| *P. palustris* | 0.76 ± 0.01 | −0.0027 ± 0.0001 | 1728 ± 88 |
| *N. sylvatica* | 0.52 ± 0.04 | −0.0064 ± 0.0007 | 721 ± 72 |
| Fire suppressed | | | |
| *A. stricta* | 0.80 ± 0.01 | −0.0021 ± 0.0001 | 2174 ± 132 |
| *P. palustris* | 0.76 ± 0.01 | −0.0027 ± 0.0002 | 1713 ± 97 |
| *N. sylvatica* | 0.51 ± 0.03 | −0.0065 ± 0.0006 | 714 ± 65 |
| **Pond 12** | | | |
| Fire maintained | | | |
| *A. stricta* | 0.79 ± 0.01 | −0.0023 ± 0.0001 | 2017 ± 75 |
| *P. palustris* | 0.75 ± 0.01 | −0.0028 ± 0.0001 | 1645 ± 71 |
| *N. sylvatica* | 0.53 ± 0.02 | −0.0062 ± 0.0003 | 749 ± 34 |
| Fire suppressed | | | |
| *A. stricta* | 0.79 ± 0.01 | −0.0023 ± 0.0001 | 2022 ± 79 |
| *P. palustris* | 0.73 ± 0.02 | −0.0030 ± 0.0003 | 1535 ± 159 |
| *N. sylvatica* | 0.50 ± 0.03 | −0.0067 ± 0.0005 | 692 ± 51 |

(27.2 ± 5.3 (mean ± SE) invertebrates per gram of leaf litter remaining; Tukey's HSD: $t_{45}$ = 1.4, $P = 0.17$). Similarly, *N. sylvatica* leaf packs (17.4 ± 2.8 (mean ± SE) invertebrates per gram of leaf litter remaining) had higher flatwoods salamander prey relative abundance than *P. palustris* (4.3 ± 0.9 (mean ± SE) invertebrates per gram of leaf litter remaining) and *A. stricta* (8.8 ± 1.3 (mean ± SE) invertebrates per gram of leaf litter remaining) (Tukey's HSD: $t_{45}$ = 5.1, $P < 0.0001$ and $t_{45}$ = 3.3, $P = 0.002$, respectively) (Fig. 2).

The NMDS ordination showed only marginal separation among habitat types and larger separation among leaf types (Stress = 0.13; Fig. 3). The PERMANOVA indicated that both habitat (PERMANOVA: *pseudo-*$F_{1,49}$ = 4.30, $P = 0.003$) and leaf litter type (PERMANOVA: *pseudo-*$F_{2,49}$ = 3.62, $P = 0.001$) were significantly affecting invertebrate

**Table 2** **Parameter estimates for three linear mixed effects models.** Linear mixed effects models estimated the effects of habitat type (fire suppressed or fire maintained) and leaf litter species (Longleaf Pine (*Pinus palustris*), Pineland Threeawn Grass (*Aristida stricta*), and Black Gum (*Nyssa sylvatica*)) on leaf litter breakdown, invertebrate relative abundance, and the relative abundance of invertebrate groups that are important prey items for larval flatwoods salamanders. Leaf litter breakdown and invertebrate communities were measured using leaf packs in three wetlands on Eglin Air Force Base, Florida.

| Model | Habitat | Leaf species | Estimate | Standard error |
|---|---|---|---|---|
| Leaf litter breakdown | Fire suppressed | *N. sylvatica* | 8.21 | 0.18 |
| | Fire Suppressed | *P. palustris* | 3.13 | 0.20 |
| | Fire Suppressed | *A. stricta* | 3.83 | 0.21 |
| | Fire maintained | – | −0.48 | 0.20 |
| Invertebrate relative abundance | Fire suppressed | *N. sylvatica* | 31.53 | 10.17 |
| | Fire suppressed | *P. palustris* | −18.12 | 12.70 |
| | Fire suppressed | *A. stricta* | −4.14 | 13.10 |
| | Fire maintained | – | 16.73 | 12.70 |
| Relative larval flatwoods salamander prey abundance | Fire suppressed | *N. sylvatica* | 18.78 | 3.00 |
| | Fire suppressed | *P. palustris* | −14.16 | 3.63 |
| | Fire suppressed | *A. stricta* | −9.80 | 3.74 |
| | Fire maintained | – | −2.82 | 3.63 |

**Table 3** **Invertebrate abundance in leaf packs that were placed in pine flatwoods wetlands on Eglin Air Force Base, Florida.** Invertebrate groups collected in 54 leaf litter packs submerged for 103–104 days in ephemeral wetlands. Leaf packs containing either Longleaf Pine (*Pinus palustris*), Pineland Threeawn Grass (*Aristida stricta*), or Black Gum (*Nyssa sylvatica*) were placed in either fire-suppressed or fire-maintained habitat in three wetlands. Values represent the mean (±standard deviation) abundance of each invertebrate group averaged across nine replicates. Taxonomic groups marked with an asterisk were dominated by a single family but were grouped to higher taxonomic levels for all analyses (Isopoda - Asellidae, Amphipoda - Gammaridae, Diptera - Chironomidae, Hemiptera - Corixidae, Zygoptera - Coenagrionidae, Acariformes - Hydrachnidae, Gastropoda - Planorbidae).

| | Fire suppressed | | | Fire maintained | | |
|---|---|---|---|---|---|---|
| | Gum | Pine | Wiregrass | Gum | Pine | Wiregrass |
| Isopoda* | 129.3 ± 113.3 | 41.4 ± 52.1 | 80.4 ± 70.6 | 105.0 ± 40.5 | 42.6 ± 36.1 | 91.7 ± 57.7 |
| Amphipoda* | 8.6 ± 4.0 | 9.4 ± 8.6 | 13.8 ± 26.6 | 12.6 ± 23.3 | 2.0 ± 2.7 | 5.4 ± 5.5 |
| Anostraca | 0.1 ± 0.3 | — | — | — | — | — |
| Cladocera | 23.0 ± 23.7 | 9.6 ± 7.3 | 62.1 ± 81.2 | 189.8 ± 329.1 | 64.0 ± 93.9 | 127.8 ± 182.0 |
| Copepoda | 7.7 ± 5.7 | 1.3 ± 1.4 | 5.7 ± 4.3 | 4.0 ± 4.2 | 1.0 ± 1.3 | 1.2 ± 1.7 |
| Decapoda | — | — | — | 0.2 ± 0.4 | — | — |
| Diptera* | 38.2 ± 13.7 | 24.7 ± 20.8 | 22.6 ± 17.9 | 12.3 ± 5.3 | 9.7 ± 11.2 | 12.1 ± 14.0 |
| Hemiptera* | — | 0.2 ± 0.4 | — | — | 0.1 ± 0.3 | 0.2 ± 0.4 |
| Coleoptera | 0.2 ± 0.4 | 0.1 ± 0.3 | — | 0.1 ± 0.3 | 0.1 ± 0.3 | 1.6 ± 2.3 |
| Odonata | — | 0.1 ± 0.3 | — | — | — | — |
| Anisoptera | — | — | — | 0.6 ± 1.3 | 0.2 ± 0.7 | 0.1 ± 0.3 |
| Zygoptera* | — | — | — | 1.0 ± 1.4 | 1.6 ± 2.4 | 2.7 ± 3.8 |
| Acariformes* | 38.3 ± 69.7 | 63.2 ± 96.7 | 122.8 ± 170.0 | 21.4 ± 25.2 | 56.8 ± 64.4 | 69.2 ± 67.9 |
| Collembola | 0.3 ± 0.7 | 0.0 ± 0.0 | 0.3 ± 1.0 | 0.2 ± 0.4 | 0.1 ± 0.3 | — |
| Gastropoda* | 1.2 ± 2.6 | 0.7 ± 1.3 | 0.1 ± 0.3 | 14.4 ± 20.3 | 2.8 ± 3.3 | 2.3 ± 4.2 |
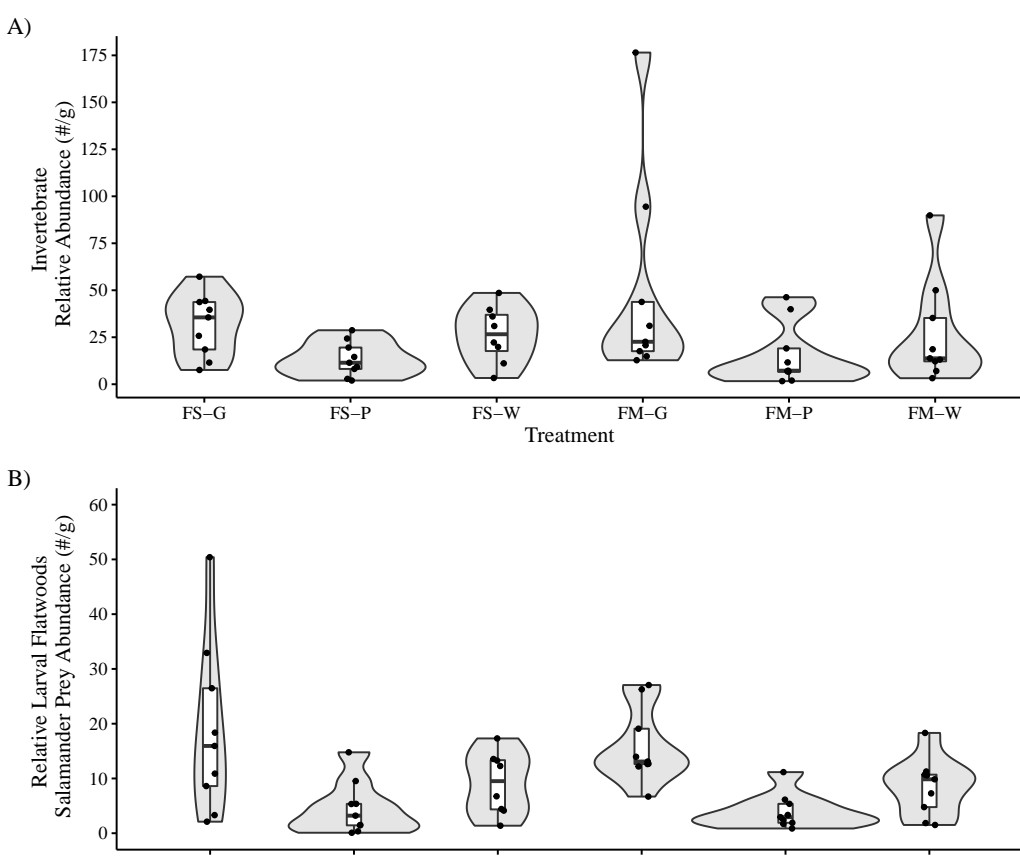

**Figure 2** **Invertebrate relative abundance measured in leaf packs in three pine flatwoods wetlands on Eglin Air Force Base, Florida.** (A) Total relative invertebrate abundance. (B) Relative abundance of taxa (Isopoda, Amphipoda, and Copepoda) important to larval flatwoods salamander diets. Leaf packs represented one of six treatments that varied by leaf type and the overall habitat type in that portion of the wetland (FS-G: fire-suppressed, Black Gum (*Nyssa sylvatica*); FS-P: fire-suppressed, Longleaf Pine (*Pinus palustris*); FS-W: fire-suppressed, Pineland Threeawn Grass, commonly referred to as wiregrass, (*Aristida stricta*); FM-G: fire-maintained, *N. sylvatica*; FM-P: fire-maintained, *P. palustris*; FM-W: fire-maintained, *A. stricta*). Relative abundance is calculated as the number of invertebrates divided by the dry mass of remaining leaf litter, and polygons represent the mirrored kernel density plot, showing the smoothed distribution of the data points.

community composition. Similar to the above results, invertebrate communities observed on *N. sylvatica* had larger differences from those observed on *P. palustris* and *A. stricta* leaf litter, while the community observed on *P. palustris* almost completely overlapped the community observed on *A. stricta* (Fig. 3). The distance-based tests for homogeneity of community dispersions indicated that variation was similar across groups (Leaf: *pseudo-$F_{2,50}=0.1.6$, $P=0.23$*; Habitat: *pseudo-$F_{1,51}=0.98$, $P=0.34$*).

Environmental characteristics varied across the fire-maintained and fire-suppressed sections of wetlands. Herbaceous vegetation cover was nearly absent from fire-suppressed sections of the three wetlands ($2.5 \pm 0.0\%$ in each wetland (mean $\pm$ SE)), while canopy cover showed more variation across the three wetlands (33–92%). In the fire-maintained

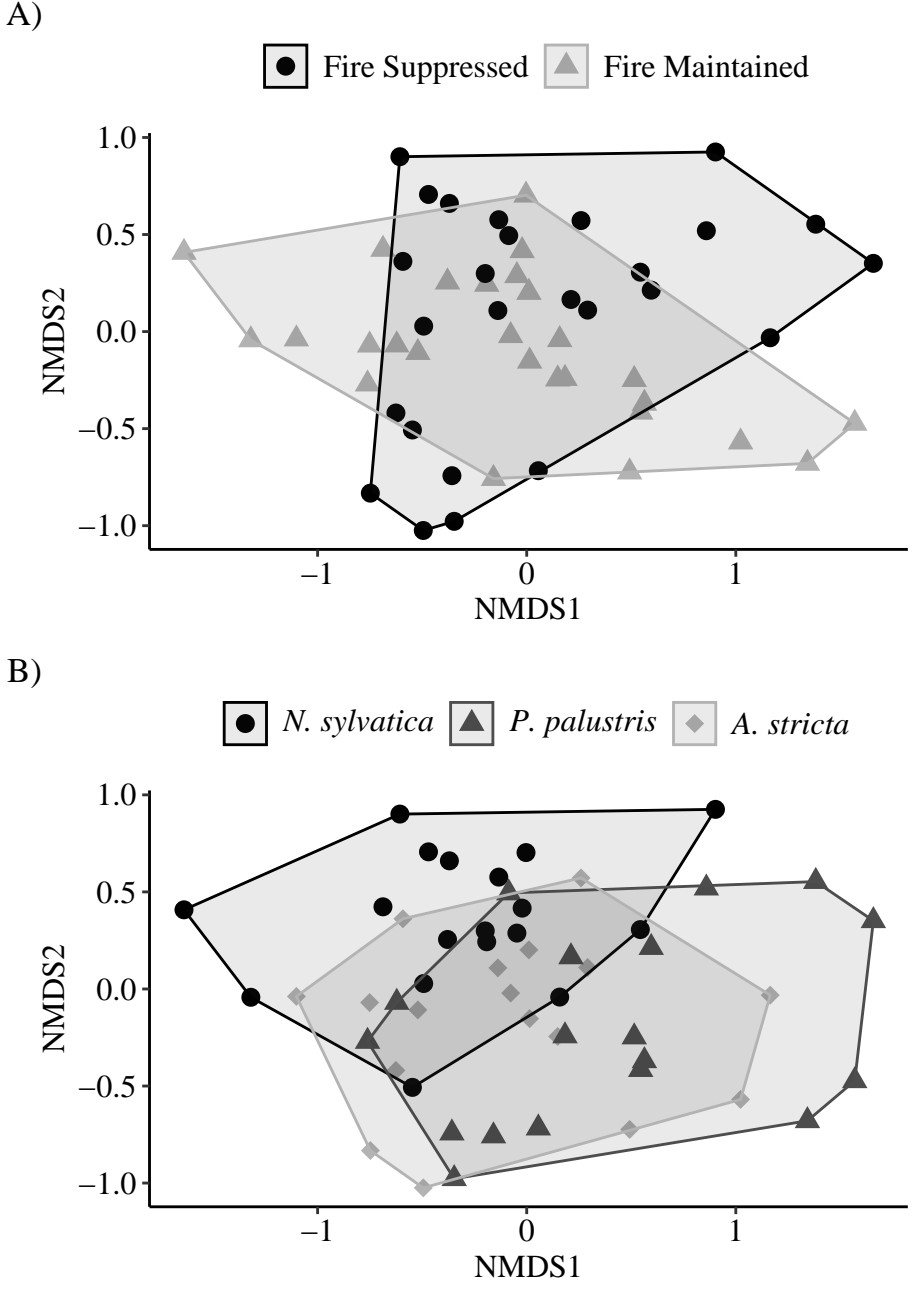

**Figure 3** **Non-metric Multidimensional Scaling (NMDS) plots of invertebrate communities measured in leaf packs from pine flatwoods wetlands on Eglin Air Force Base, Florida.** (A) Leaf packs were placed in either fire-suppressed or fire-maintained habitat. (B) Leaf packs contained one of three leaf types: Longleaf Pine (*Pinus palustris*), Pineland Threeawn Grass (*Aristida stricta*), or Black Gum (*Nyssa sylvatica*).

habitat, herbaceous vegetation cover ranged from 63–85%, and canopy cover ranged from 8–16%.

## DISCUSSION

The ecology of pine flatwoods wetlands is primarily regulated by wetland hydrology and fire regime. Both of these factors directly impact floral and faunal communities within wetlands along with a variety of biotic and abiotic processes that shape wetland ecosystems (*Brockway & Lewis, 1997*; *Powell et al., 2005*; *Gorman, Haas & Bishop, 2009*; *Watts, 2013*). Our results indicate that vegetation community composition within pine flatwoods wetlands impacts both leaf litter breakdown and aquatic invertebrate communities. Overall, leaf species had a larger effect on these processes than the broad vegetation characteristics within a wetland. However, in natural wetlands, leaf species and vegetation characteristics are fundamentally linked because certain tree and shrub species (*i.e.,* ones that increase canopy cover) are generally limited to either fire-suppressed sections of wetlands or the deepest areas in fire-maintained wetlands (*Kirkman, 1995*). Our results, along with other work, strongly indicate that altered fire regimes (particularly a loss of growing-season fire) fundamentally alter wetland ecosystems that are embedded within fire-dependent landscapes (*De Szalay & Resh, 1997*; *Bixby et al., 2015*; *Chandler, Haas & Gorman, 2015*).

Over the course of our study, a majority of the observed mass loss in all three leaf species occurred within 25 days of leaf packs being submerged. This rapid initial breakdown is generally driven by a combination of microbial activity and leaching of secondary compounds into the aquatic environment (*Tietema & Wessel, 1993*). After this initial period, *N. sylvatica* leaves continued to break down, while both *P. palustris* and *A. stricta* leaf packs lost little mass after the initial period. Over the entire study period, *N. sylvatica* leaf packs lost approximately 15–28% more mass than either *P. palustris* or *A. stricta*. The decay rates that we observed for *N. sylvatica* are similar to those previously reported (*Battle & Golladay, 2001*; *Battle & Golladay, 2007*). Interspecific differences in leaf litter breakdown can be attributed to the chemical make-up of the leaves (*Battle & Golladay, 2007*) and the abundance of invertebrates, particularly shredders, present (*Tiegs et al., 2008*), although the effects of invertebrates on leaf breakdown are variable across studies (*Battle & Golladay, 2001*; *Fuell et al., 2013*). Our results showed that *N. sylvatica* leaf packs contained more invertebrates relative to the amount of material remaining, possibly contributing to their continued breakdown across the entire 104-day period.

We also observed differences in breakdown rates between habitat types, although these effects were smaller overall and appeared to vary among the three wetlands. Leaf litter breakdown in ephemeral wetlands is driven by multiple processes that span both aquatic and terrestrial environments, such as water temperature, water chemistry, flooding regime (both depth and hydroperiod), and the above-mentioned biotic factors (*Álvarez & Bécares, 2006*; *Battle & Golladay, 2007*; *Gingerich, Panaccione & Anderson, 2015*). Canopy cover was variable across the three wetlands included in this study, potentially influencing the breakdown rate between habitat types in individual wetlands by impacting water temperatures (*Werner & Glennemeier, 1999*; *Becker et al., 2012*). This interpretation is
supported by our observation that the largest within-wetland difference in breakdown rates occurred in the wetland with the highest canopy cover in the fire-suppressed section (*i.e.,* 92% in Pond 53 compared to just 62% and 33% in Ponds 12 and 16, respectively; Table 1). All fire suppressed sections of wetlands had almost no herbaceous vegetation but the variability in canopy cover may reflect differences in the severity of fire suppression or in the species composition of the mid-story vegetation (*Peterson & Reich, 2008*). While none of the wetlands experienced fire during our study, fires can impact water quality within wetlands, potentially altering breakdown rates (*Battle & Golladay, 2003*). Overall, it appears that differences between fire-suppressed and fire-maintained habitats may alter leaf breakdown rates in some cases, but that these effects are inconsistent and small compared to differences between leaf litter types.

Observed differences in invertebrate relative abundance were attributed to the different leaf litter species but not to the overall habitat type. Across both habitat types, *N. sylvatica* leaf packs tended to support more abundant invertebrate communities relative to the amount of leaf litter remaining at the end of the study. Higher invertebrate abundance may be related to differences in leaf structure for invertebrate colonization (*e.g.*, complex *N. sylvatica* leaves *vs.* cylindrical *P. palustris* needles and *A. stricta* stalks). Greater complexity and surface area can increase the abundance and diversity of invertebrate colonizers (*Beckett, Aartila & Miller, 1992*; *Jeffries, 1993*). Furthermore, leaf litter quality is often hypothesized to account for differences in invertebrate abundance across leaf species, and coniferous leaf litter has lower nutrient quality when compared to deciduous leaf litter (*Polyakova & Billor, 2007*; *Hisabae, Sone & Inoue, 2010*). However, coniferous leaf litter can be an important food resource for invertebrate communities when other litter types are scarce (*Sakai et al., 2016*), which may be the case in fire-maintained pine flatwoods wetlands where deciduous leaf litter is generally absent.

Our results indicated that there was little effect of overall habitat type on invertebrate relative abundance when sampling using leaf packs. These results starkly contrast some of our previous survey work in these wetlands that used dip net surveys to quantify invertebrate communities and showed a large difference across fire-maintained and fire-suppressed habitats (*Chandler et al., 2015*). These observed differences between studies are likely driven by the different sampling methodologies employed (*i.e.,* sampling in the water column *vs.* colonization in leaf packs). Ultimately, both leaf packs and dip net surveys over relatively small temporal and spatial scales are only targeting a subset of the aquatic invertebrate communities in these wetlands. In tandem, the results reported here and from *Chandler et al. (2015)* indicate that invertebrate abundance is likely similar throughout wetland habitat types but that both the composition and spatial arrangement (*i.e.,* in leaf litter *vs.* in standing herbaceous vegetation) of invertebrate communities varies across habitats.

Differences in the habitat use of invertebrate communities could have important implications for flatwoods salamanders because larvae are primarily found in areas with thick herbaceous vegetation (*Sekerak, Tanner & Palis, 1996*; *Gorman, Haas & Bishop, 2009*), and wetlands lacking sufficient area with dense herbaceous vegetation support few or no flatwoods salamanders (*Brooks et al., 2019*; *Wendt et al., 2021*). This association may be linked to egg deposition sites (*Gorman et al., 2014*) but could also reduce predation

pressure, increase foraging opportunities for larvae, or both. Limited observations suggest that larval flatwoods salamanders forage in and around herbaceous vegetation and along edges of wetland basins (*Palis, 1996*; *Sekerak, Tanner & Palis, 1996*; *Whiles et al., 2004*), but larvae can be forced into deeper areas with high canopy cover and little herbaceous vegetation when wetland drying occurs (*Chandler et al., 2017*). Of the three most common invertebrate groups found in larval flatwoods salamander stomachs, Isopoda was by far the most abundant in leaf packs, regardless of litter species or habitat type, suggesting that isopods are a plentiful food source in most wetlands. Even though there were no differences in the overall abundance of the three most abundant invertebrate groups predated by flatwoods salamander larvae across habitat types, Cladocera abundance was higher in fire-maintained sections of wetlands, and these small crustaceans are an important part of the diets of small salamander larvae (*Whiles et al., 2004*). Overall, during a typical cycle of wetland flooding, invertebrate abundance appears unlikely to be a limiting factor for flatwoods salamander larvae but both hydroperiod and the timing of flooding could significantly impact invertebrate community composition and abundance (*Gleason & Rooney, 2017*).

Compared to many lotic systems, the available information on nutrient cycling, leaf breakdown, and invertebrate communities in ephemeral wetlands embedded within Longleaf Pine forests is minimal. Our study was limited in scope, and future studies could expand on this work in several ways. First, to acquire a more comprehensive understanding of this ecosystem, a complete inventory of aquatic invertebrate species could be conducted in these wetlands, focusing on identifying invertebrates to lower taxonomic levels. Higher taxonomic resolution would allow for a better assessment of the role that specific invertebrates fill in wetland food webs (*e.g.*, *Golladay, Taylor & Palik, 1997*). Second, the shrub layers that develop in fire-suppressed wetlands are diverse, both in terms of species composition and leaf characteristics. Other common shrub species (*e.g.*, members of the genus *Ilex*) may have different effects on invertebrate communities than those observed in this study. Furthermore, examining the chemical composition of common litter types would shed light on the causal relationships between litter quality and invertebrate abundance. Third, there is a paucity of data on water quality and broader nutrient cycling in pine flatwoods wetlands (but see *Sun et al., 2006*). Finally, our study only examined leaf breakdown and invertebrate communities across a continuously flooded time period. However, hydrology, especially wetland drying, can have significant effects on these processes (*Battle & Golladay, 2007*; *Gleason & Rooney, 2017*), and vegetation community composition can in turn impact wetland hydrology (*McLaughlin, Kaplan & Cohen, 2013*; *Golladay et al., 2021*).

## CONCLUSIONS

Our study adds to the large body of literature demonstrating the effects of historic and contemporary fire suppression and exclusion on pine ecosystems in the southeastern United States. Our results indicated that vegetation changes associated with a lack of growing-season fires can alter both breakdown and invertebrate communities in wetland systems.

Leaf litter inputs into wetlands form the foundation of aquatic food webs and contribute to the overall cycling of nutrients within the wetland. These processes are also linked to the surrounding uplands through annual movements of animals into and out of wetlands in response to flooding events (*Smith et al., 2019*). Ultimately, management of pine flatwoods wetlands should prioritize maintaining or restoring vegetation structure characteristic of a fire-dependent ecosystem through a combination of mechanical treatments and prescribed fire applied during the growing season or most importantly when wetlands are dry, allowing fire to carry through wetland basins (*Gorman, Haas & Himes, 2013*).

## ACKNOWLEDGEMENTS

We thank the many people who assisted with study design, fieldwork, and sample processing, especially B Chandler, R Chandler, K Jones, M Cawthorn, B Collins, M McKeon, B Rincon, and T Williams. The manuscript was improved by comments from two anonymous reviewers. Logistical support was provided by Jackson Guard (Eglin Air Force Base's Natural Resources Division), the Virginia Tech Department of Fish and Wildlife Conservation, and the Georgia Southern University Department of Biology.

### Funding

Funding was provided by Jackson Guard (Eglin Air Force Base's Natural Resources Division), the Virginia Tech Department of Fish and Wildlife Conservation, and the Georgia Southern University Department of Biology. Early investment in the larger scale research project on habitat conditions and the effects of invertebrates important to flatwoods salamanders at Eglin Air Force Base was provided by Hurlburt Field through a wetland mitigation project, the Florida Fish and Wildlife Conservation Commission Aquatic Habitat Restoration and Enhancement Project, Department of Defense Legacy Project 12-109. This work was also supported by the USDA National Institute of Food and Agriculture, McIntire Stennis project 1024640. The funders had no role in study design, data collection and analysis, decision to publish, or preparation of the manuscript.

### Grant Disclosures

The following grant information was disclosed by the authors:
Jackson Guard (Eglin Air Force Base's Natural Resources Division).
The Virginia Tech Department of Fish and Wildlife Conservation.
The Georgia Southern University Department of Biology.
Hurlburt Field through a wetland mitigation project.
The Florida Fish and Wildlife Conservation Commission Aquatic Habitat Restoration and Enhancement Project.
Department of Defense Legacy Project 12-109.
The USDA National Institute of Food and Agriculture, McIntire Stennis project 1024640.

## Competing Interests

The authors declare there are no competing interests.

## Author Contributions

- Houston C. Chandler conceived and designed the experiments, performed the experiments, analyzed the data, prepared figures and/or tables, authored or reviewed drafts of the paper, and approved the final draft.
- J. Checo Colón-Gaud, Thomas A. Gorman and Carola A. Haas conceived and designed the experiments, authored or reviewed drafts of the paper, and approved the final draft.
- Khalil Carson performed the experiments, authored or reviewed drafts of the paper, and approved the final draft.

## Field Study Permissions

The following information was supplied relating to field study approvals (i.e., approving body and any reference numbers):

Field experiments and access to field sites were approved by the U.S. Fish and Wildlife Service and Jackson Guard (Eglin Air Force Bases Natural Resources Division) (Cooperative Agreement Number F14AC00068). No permits are required to work with non-listed invertebrates in Florida.

## Data Availability

The raw data are available in the Supplemental Files.

## Supplemental Information

Supplemental information for this article can be found online at http://dx.doi.org/10.7717/peerj.12534#supplemental-information.

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
