# Peer review of "Does long-term fire suppression impact leaf litter breakdown and aquatic invertebrate colonization in pine flatwoods wetlands?"

_PeerJ, doi:10.7717/peerj.12534_

## Round 0.1 · original submission · Major Revisions

Dear Chandler et al.,

After the consideration by two independent reviewers, I consider your manuscript may be published in PeerJ after you perform improvements to it. Please pay special attention to all the issues raised by the reviewers. Considering the significant changes, you will need to make. I believe a two-month period (November 15th, 2021) is enough for you to improve your study. In case you need more time, please let me know. Do not hesitate to resubmit earlier if you can.

By the time of resubmission, do not forget to prepare a rebuttal letter informing all the performed changes and justifying any of those suggestions which you disagreed with the issues raised by the reviewers.

Sincerely,
Daniel Silva, PhD

Reviewer 1 ·

Basic reporting

The article is well written with a few places with excessive wordiness. The literature cited has some problems.

Lines 183-189: This is a lot of text to simply state that one litter bag was removed due to concerns about leaf mass loss probably caused by externalities. Condense.

Lines 433-435, 468, 505, 513: Please ensure literature cited is properly formatted.

Experimental design

This manuscript has many positive aspects and provides more evidence for the importance of fire in southeastern USA wetlands. The design appears to be appropriate and with clarification, the analysis is adequate. Most conclusions are well supported.

This manuscript has weaknesses in undescribed variability within the fire-maintained category, connections to salamanders, identification of insects to order, and the lack of reporting any measures of statistical variability. Fire prescriptions and applications can vary intentionally (e.g., different regimes for different outcomes) and circumstantially (e.g., poor weather). Thus, there could be more variability within this category than between fire-maintained and fire-suppressed. To the best of their ability, the authors need to describe the fire prescription (frequency at least) and actual application. Although all fire is occurring on one installation by one agency, one cannot assume that the actual fire-maintenance is the same across study sites. If the study is to be connected to salamanders, more information is needed to explain how these data inform salamander conservation. At a minimum, the authors need to provide more information on prey items and foraging by the salamanders to make the connection. As the authors acknowledge, identification to order limits the study. Interestingly, the community results were among the strongest. Potentially, identification to family or genus would have enhanced these results further.

Methods:
Lines 141-142: You need to provide more details on fire-maintained. Were all of these fires at similar times of year? Did they occur at similar frequencies? There could be a lot of variability in the application of prescribed fire. It is understandable that local conditions and weather can make each fire different, however, if the prescription is similar, this is sufficient as variability within the category would be due to local conditions. If the prescriptions varied, the variability within fire maintained may be too great to be a meaningful category.

Lines 164-165: There are multiple formulae and methods in the literature for estimating k. Indicate here the method that you used. Did you use the LMM for this as well? The manuscript reads as if estimation of k was one step and the LMM was another step, however, it also could be interpreted to be the same. If these were the same, why is k depicted as a nonlinear relationship in Figure 1? The estimation method for k needs to be provided to resolve this issue. If the estimation of k was separate from the LMM, why not estimate k with litter type and habitat type variables in one analysis? Generalized linear models and nonlinear models allow for exponential decay estimation with other variables included.

Line 167: Order is a very coarse measure. This is a real limitation in the manuscript.

Lines 190-202: Although the LMM is a very useful and powerful analysis, the LMM does have a number of important assumptions. Please describe how these assumptions (e.g., linearity, normality of residuals, appropriate estimation of the G matrix) were evaluated and whether alternatives, such as generalized linear mixed models were evaluated. Given experience, these data were likely better analyzed with a generalized linear mixed model or generalized hierarchical mode. Abundance data very rarely meet the assumptions of LMM. Moreover, k is a rate with the same concerns. Abundances and rates are often better modeled by Poisson and negative binomial models with the generalized linear model framework. Abundances and k may have met LMM assumptions here, but it is the authors responsibility to establish this.

Lines 203-213: This is a well described and very appropriate analysis. Like you did in lines 210-211, you should describe how you evaluated the appropriate use of LMM in Lines 190-202.

Lines 236-243: First, mention the use of a priori contrasts in the Methods. The first mention of a priori contrasts should not be the Results. Second, again, standard errors or 95% confidence intervals are needed. 4.3 sounds impressively less than 8.7, but 4.3±2.0SE is not really different than 8.7±3.0SE.

Validity of the findings

Throughout the manuscript, measures of statistical variability are not reported. Comparing means without standard errors or 95% confidence intervals is not reporting all relevant information and not giving context. A mean of 10 seems larger than a mean of 5, except that a mean of 10 with a 95% confidence interval of 0-20 may not be meaningfully different than a mean of 5 with a 95% confidence interval of 0-15. These measures matter and must be reported for the reader to fully interpret your data, interpretations, and conclusions.

Abstract: Please add standard errors or 95% confidence intervals to K, invertebrates per pack, and invertebrates per gram. Explain how community composition differed. This is more important than methods details.

Results:
Lines 218-225: First, please present the results in the same order as the methods. Therefore, present the estimates for k first. Second, please add parameter estimates (betas) and their standard errors or 95% confidence intervals. These greatly aid in understanding the relative contribution of each variable. Consider standardized effect sizes to assist interpretation. Third, add standard errors or confidence intervals to k.

Lines 226-230: Standard errors or 95% confidence intervals are more helpful than ranges.

Lines 230-236: Similar to previous comment, parameter estimates and standard errors are very helpful.

Lines 236-243: First, mention the use of a priori contrasts in the Methods. The first mention of a priori contrasts should not be the Results. Second, again, standard errors or 95% confidence intervals are needed. 4.3 sounds impressively less than 8.7, but 4.3±2.0SE is not really different than 8.7±3.0SE.

Lines 256: Present standard error or 95% confidence intervals for this mean.

Discussion:
Lines 296-297: The variability among wetlands would be easier to understand for the reader, if confidence intervals were provided for k in the text and in Figure 1.

Lines 305-307: These data are not presented in a comparable manner. If you are going to make the point that canopy cover is important, the relevant canopy cover and invertebrate data need to be presented together in text or table.

Lines 326-337: Comparing invertebrate samples between an active and passive sampler is very difficult. This is why comprehensive invertebrate studies include multiple sampling devices. There is nothing wrong with using one method, especially if the method is the best option for the research question. Waterfowl researchers often only sample surface or open water invertebrates, because these are the relevant organisms. Here, you sampled using leaf packs, which is where salamanders presumably forage, although this could be clarified in the manuscript. If indeed the leaf packs, represent where the salamanders forage and allowed for concurrent estimation of leaf decay, you can state this and shorten this paragraph to: 1) we found this; 2) Chandler et al. (2015) found that; 3) we used different methods relevant to our question; and 4) continue with your conclusion.

Lines 338-342: There are two ideas in this paragraph that are not well connected. First, an argument is made that salamander larvae need herbaceous cover and the Results do report very little herbaceous cover in fire suppressed wetlands. Second, prey requirements are discussed, presumably in relation to herbaceous cover differences (if so, make this clearer). Isopods are leaf litter associates but Cladocera are not. Therefore, it is not clear where in the wetland that the salamanders forage. Here and in the Introduction, you need to describe where in the wetland salamanders forage, as this matters to your sampling method and to your interpretation of comparisons across treatments, and what are the top forage items, as this puts context to whether Isopods matter. If salamanders are not foraging in leaf litter, you really need to drop the connection to salamanders in this study.

Also, considering the previous comment on Lines 326-337, did you really adequately sample Cladocera?

Additional comments

Introduction:
Lines 61-62: Wildlife should be extensive, but is suppressed. Therefore, pervasive is not the right way to word it.
Lines 68-71: Except where actively managed by fire. It is important to note this.
Somewhere in the Introduction, you need to connect the salamanders to these wetlands and invertebrates to better support your later analyses and discussion.

Reviewer 2 ·

Basic reporting

The article is well written and professional english used throughout.

A few additional recent citations were provided but the background/context is very good.

Article structure is fine and raw data are provided.

The results directly address study objectives.

Experimental design

This manuscript fits well within the aims and scope of PeerJ

The authors have identified and interesting and important research area/knowledge gap.

Technical implementation of the study is well explained and subscribes to accepted standards.

A few suggestions were made on the manuscript to improve details, but generally methods were adequate to allow replication

Validity of the findings

There was not a large effect of treatment (fire versus fire suppression) as the authors expected. However, this is interesting because it suggests that there is a great deal that we still don't understand about isolated wetland foodwebs in longleaf forest. I have challenged the authors to think a bit more about their results and what they mean in the context of how we manage wetlands in fire-maintained forests.

I have requested a bit more detail on data analysis, but the overall approach is statistically sound,

The discussion is a bit long in places, but I have provided a suggestion or two for condensation.

None of these comments undermine the value of this research and its presentation for publication. Standards for the journal appear to have been met.

Additional comments

I made some comments about clarifying the name of the wetlands studied in this manuscript. I believe that these suggestions will help readers visualize the study system. The authors and editors are free to reject these suggestions.

The authors are measuring leaf breakdown or leaf processing, which is not the same as leaf decomposition. I recommend adopting the breakdown/processing terminology through out the manuscript.

I made suggestions on the text of the manuscript and it is provided. I hope the authors find my suggestions constructive as that is how I intend them. I feed privileged to read a preview version of this study.

Annotated reviews are not available for download in order to protect the identity of reviewers who chose to remain anonymous.

---

## Round 0.2 · accepted · Accept

Dear Dr. Chandler,

I am pleased to inform you that your manuscript has been accepted for publication in PeerJ. Congratulations!

Best regards,
Daniel Silva

Reviewer 1 ·

Basic reporting

This revision is greatly improved.

Experimental design

Clarifications in this revision enhanced the manuscript.

Validity of the findings

Revisions were successful in conveying the validity of findings.

Additional comments

The authors did a very good job of revising and clarifying this manuscript.

Reviewer 2 ·

Basic reporting

The article is well written and professional english used throughout.

The background/context is very good.

Article structure is fine and raw data are provided.

The results directly address study objectives.

Experimental design

This manuscript fits well within the aims and scope of PeerJ

The authors have identified and interesting and important research area/knowledge gap.

Technical implementation of the study is well explained and subscribes to accepted standards.

Validity of the findings

The findings are consistent with the study design and data presented. I see no issues with validity

Additional comments

I previously reviewed this manuscript. The authors have done an excellent job of addressing my questions and concerns. I believe that review cycles should have an end, so I do not have additional comments on this manuscript.